# Persistence of Tembusu Virus in *Culex tritaeniorhynchus* in Yunnan Province, China

**DOI:** 10.3390/pathogens12030490

**Published:** 2023-03-21

**Authors:** Danhe Hu, Chao Wu, Ruichen Wang, Xiaohui Yao, Kai Nie, Quan Lv, Shihong Fu, Qikai Yin, Wenzhe Su, Fan Li, Songtao Xu, Ying He, Guodong Liang, Xiangdong Li, Huanyu Wang

**Affiliations:** 1Jiangsu Co-Innovation Center for Prevention and Control of Important Animal Infectious Diseases and Zoonoses, College of Veterinary Medicine, Yangzhou University, Yangzhou 225009, China; 2State Key Laboratory of Infectious Disease Prevention and Control, National Institute for Viral Disease Control and Prevention, Chinese Center for Disease Control and Prevention, Beijing 102206, China; 3Yunnan Institute of Parasitic Diseases, Pu’er 665000, China; 4Guangzhou Center for Disease Control and Prevention, Guangzhou 510440, China; 5Key Laboratory of Protection & Utilization of Biological Resources in Tarim Basin, College of Life Sciences, Tarim University, Alar 843399, China

**Keywords:** Tembusu virus, mosquito-borne flaviviruses, sequence analysis, phylogeny

## Abstract

The Tembusu virus (TMUV), a member of the Flaviviridae family, can be transmitted via mosquitoes and cause poultry disease. In 2020, a strain of TMUV (YN2020-20) was isolated from mosquito samples collected in Yunnan province, China. In vitro experiments showed that TMUV-YN2020-20 produced a significant cytopathic effect (CPE) in BHK, DF-1, and VERO cells, while the CPE in C6/36 cells was not significant. Phylogenetic analysis revealed that the strain belonged to Cluster 3.2 and was closely related to the Yunnan mosquito-derived isolates obtained in 2012 and the Shandong avian-derived isolate obtained in 2014. Notably, TMUV-YN2020-20 developed five novel mutations (E-V358I, NS1-Y/F/I113L, NS4A-T/A89V, NS4B-D/E/N/C22S, and NS5-E638G) at loci that were relatively conserved previously. The results of this study demonstrate the continuous circulation and unique evolution of TMUV in mosquitoes in Yunnan province and suggest that appropriate surveillance should be taken.

## 1. Introduction

The Tembusu virus (TMUV), which is a single, positive-stranded RNA virus, is a member of the Flaviviridae family. In Malaysia, *Culex tritaeniorhynchus* was reported as the first source of TMUV in 1955 [1]. Since then, other *Culex* species, including *Culex gelidus* and *Culex vishnui*, have also been found to carry the virus [2,3]. In 2000, researchers found TMUV could infect vertebrates and cause avian disease. TMUV was isolated from samples of sick chickens from a broiler flock in Malaysia and characterized by symptoms such as encephalitis and growth retardation [4].

In 2010, TMUV was found to spread among ducks in southeast China and caused a substantial economic effect [5]. The farming sector was seriously threatened when similar epidemics were discovered in duck populations in Southeast Asia, notably Malaysia and Thailand [6,7]. With a morbidity rate of 90% to 100% and a mortality rate between 5% and 30% in infected duck flocks, TMUV infection can have clinical indications in ducks, such as growth retardation, loss of appetite, paralysis, and decreased egg production [5,8,9]. TMUV not only affects poultry (such as chickens, ducks, and geese), but also infects birds (such as pigeons and sparrows) [10,11,12] with similar clinical symptoms.

Mosquito-borne flaviviruses, such as Japanese encephalitis virus (JEV), dengue virus (DENV), and Zika virus (ZIKV), can infect humans and cause large-scale disease outbreaks [13,14]; nevertheless, no cases of TMUV infection in humans or other mammals have been reported. However, some studies have shown that intracerebral inoculation of BALB/c mice with TMUV causes severe neurological symptoms and even death [15]. Furthermore, epidemiological studies have shown that IgG antibodies or neutralizing antibodies to TMUV have been detected in the sera of some healthy people in China and Thailand [16,17]. These findings imply that TMUV can cause considerable economic losses in the poultry farming industry and pose a potential threat to public safety.

The TMUV genome contains 11,000 nucleotides that encode 10 proteins, including 3 structural proteins: capsid (C), pre-membrane (prM), and envelope (E); and 7 nonstructural (NS) proteins: NS1, NS2A, NS2B, NS3, NS4A, NS4B, and NS5 [18]. The E protein is frequently used for phylogenetic analysis of genetic variation and molecular epidemiology as it is the main surface protein of the viral particle. It also plays a crucial role in mediating virus receptor binding and virus cell membrane fusion and frequently undergoes genetic mutations under the evolutionary pressure of the host immune system [19]. TMUV strains are currently divided into 3 clusters (Cluster 1, 2, and 3) based on phylogenetic analysis of the open reading frame (ORF) region and the E gene [20,21]. Cluster 2 is currently the dominant cluster in China and Cluster 3 encompasses the majority of mosquito-derived isolates [22].

Yunnan province is located at the southwest border of China, where a considerable number of arboviruses have been isolated [23]. To monitor arboviruses in Yunnan province, we collected mosquitoes from July to August 2020 in Yunnan province and successfully isolated a strain of TMUV from *Culex tritaeniorhynchus*. The new strain belongs to Cluster 3.2 and has five novel amino acid mutations. Our findings reveal that TMUV is still spreading in mosquitoes in Yunnan province.

## 2. Materials and Methods

### 2.1. Mosquito Collection

The mosquito samples were collected from July to August 2020 in Nansa village, Yuanjiang county, Yuxi city, Yunnan province using UV mosquito traps (Kung Fu Xiaoshuai, 12 V, 300 mA; Wuhan Jixing Medical Technology Co., Ltd., Hubei, China) in pig and cattle pens. The traps were set between 18:00 p.m. and 6:00 a.m. The collected mosquito samples were morphologically characterized and mosquitoes of the same species were maintained in liquid nitrogen in pools of 50–100 until they were tested in the laboratory [24].

### 2.2. Cell Lines

Golden hamster kidney (BHK-21) cells and African green monkey kidney (VERO) cells were grown in minimal essential medium (MEM; Gibco, Grand Island, NY, USA) containing 10% fetal bovine serum (FBS; Gibco, Grand Island, NY, USA). Chicken fibroblasts (DF-1) cells were grown in Dulbecco’s modified Eagle medium (DMEM; Gibco, Grand Island, NY, USA) containing 10% FBS. *Aedes albopictus* egg (C6/36) cells were grown in Roswell Park Memorial Institute 1640 (RPMI 1640; Gibco, Grand Island, NY, USA), containing 10% FBS. BHK-21, VERO, and DF-1 cells were cultivated at 5% CO_2_ and at 37 °C, while C6/36 cells were cultured at 28 °C. All cell lines were kept in our laboratory.

### 2.3. Virus Isolation and RT-PCR Analysis

The collected mosquito samples were ground as previously reported [25]. The ground supernatants were inoculated into culture plates containing monolayers of BHK-21 and C6/36 cells grown at 37 °C, 5% CO_2_, and 28 °C, respectively. Cell supernatants were collected and inoculated into newly cultured BHK-21 and C6/36 cells for a third passaging after 1 observation cycle (6–7 days). The positive isolates that could produce cytopathic effects (CPEs) were collected after three generations for identification.

The total RNA was extracted from virus-infected cells according to the instructions of the QIAamp Viral RNA Mini Kit (Qiagen, Valencia, CA, USA). Reverse transcription was performed with Ready-to-Go^TM^ You-Prime First-Strand Beads (GE HealthCare, Little Chalfont, Buckinghamshire, UK) and random primers. Flavivirus genus-specific primers (F: 5′-TACCACATGATGGGAAAGAGAGAGAGAA-3′, R: 5′-GTGTCCCAGCCGGCGGTGTCATCAGC-3′) [26] were used for the RT-PCR detection, and the reaction program was set up in accordance with the instructions on the PrimeScriptTM One Step RT-PCR Kit Ver. 2 (Dye Plus) (TaKaRa Bio Inc, Shiga, Japan) reagent package. The reaction parameters were as follows: 94 °C for 5 min; denaturation at 94 °C for 30 s, annealing at 55 °C for 30 s, extension at 72 °C for 30 s for 35 cycles; final extension at 72 °C for 10 min. The intended target product size was 310 bp and the RT-PCR amplification products were sequenced using Sanger sequencing.

### 2.4. Viral Titer Determination

The virus isolate was propagated in a BHK-21 cell and a standard plaque assay was used to calculate the virus titers. Briefly, a 6-well culture plate was filled with a succession of 10-fold dilutions of the virus. Each well received 4 mL of 1.1% methylcellulose-MEM semisolid media with 2% FBS after incubation. After 4 days of incubation, we noticed the plaques clearly developing. The plaque formation units and virus titers were calculated after staining cells with crystalline violet solution.

### 2.5. Virus Growth Kinetics

As previously described, BHK-21, DF-1, VERO, and C6/36 cells were infected with the virus at a multiplicity of infection (MOI) of 0.1, respectively. After 1 h of incubation, the cells were gently washed with PBS (Invitrogen, Carlsbad, CA, USA), then the cell culture medium was added, and the cells were placed under appropriate conditions. The cell and supernatant samples were collected at 0, 12, 24, 36, 48, 60, 72, and 96 hpi and stored at −20 °C. A standard plaque assay was used to calculate viral titers.

### 2.6. Virus Genome Sequencing

For high-throughput sequencing, RNA from the flavivirus positive isolate’s third generation cell culture supernatant was used. Sequencing libraries were created using the Trio RNA-SeqTM library preparation kit guidelines (Tecan Genomics, Männedorf, Switzerland). The libraries were filtered with AMPure XP Beads (Beckman Coulter, Brea, CA, USA) and assessed for library quality with the Qubit4.0 Fluorometer (Life T Technologies, Grand Island, NY, USA). The library was then sequenced on an Illumina Novaseq 6000 platform, resulting in 150 bp paired-end reads. Offline data were filtered based on Q30 values and read lengths for further data analysis. The sequencing data were spliced using the reference genome NC015843 sequence as a template after low-quality reads were eliminated using CLC Genomics Workbench. Moreover, we sequenced the original mosquito pool using the Sanger method. Primers were acquired from previous articles (Appendix A) [27]. The sequence was submitted to NCBI (accession number: OQ238827).

### 2.7. Sequence and Phylogenetic Analysis

TMUV sequences with temporal, host, and geographic information were acquired from GenBank, and the newly sequenced sequences from this work were added to them, resulting in a dataset of 148 TMUV sequences for analysis (Appendix A). Multiple sequence alignment analysis was performed using MAFFT software (v7.450) [28]. MegAlign (DNAStar, Madison, WI, USA) was used to analyze viral sequences for nucleotide and amino acid homology [29]. A maximum likelihood (ML) tree reconstruction of 148 TMUV genome sequences was performed using IQ-TREE software (v1.6.12), and the best-fit alternative model for the TMUV genome determined by ModelFinder software was GTR + F + R3 [30]. The final tree files were visualized and annotated using the Chiplot online website (www.chiplot.online, accessed on 15 November 2022). Python scripts were used for mutation analysis in the gene sequences of the TMUV.

### 2.8. Bayesian Phylogenetic Reconstruction and Time to the Most Recent Common Ancestry (tMRCA)

The maximum clade credibility (MCC) tree was created using Beast software (v1.10.4) based on 148 full-length E-gene sequences. The alternative GTR + G + I model was employed in this investigation, with the strict clock was selected as the clock type and the Bayesian skyline as the tree prior, so Markov chain Monte Carlo (MCMC) was 200,000,000 and data gathering was conducted every 20,000 generations. To ensure that the estimated sampling size (ESS) were greater than 200, the results were analyzed visually using Tracer software (v1.7.1) [31]. The TreeAnnotator tool in the Beast package was used to process the data. The tree with the highest posterior probability (PP) with a 10% burn-in was chosen to represent a target tree. FigTree software 1.4.4 was used to visualize and comment on the final tree.

## 3. Results

### 3.1. Mosquito Collection

A total of 2723 *Culex tritaeniorhynchus* and 2273 *Anopheles sinensis* were collected from Yunnan province, China, between July and August 2020.

### 3.2. Virus Isolation and Identification

The mosquitoes were combined into pools of 50–100. A total of 94 mosquito pools were homogenized and inoculated in parallel with the BHK-21 and C6/36 cells. One pool from the BHK-21 cells and six pools from the C6/36 cells showed CPE.

RT-PCR was used to analyze positive isolates using the Flavivirus genus primers and the results showed that only the pool from the BHK-21 cells was positive. This positive isolation derived from *Culex tritaeniorhynchus* was identified as TMUV by NCBI-BLAST for the Sanger sequencing of RT-PCR production and was named TMUV-YN2020-20.

### 3.3. Virus Growth Kinetics

Mosquito (C6/36), avian (DF-1), and mammalian (BHK-21, VERO) cell lines were used to compare the susceptibility to TMUV-YN2020-20 (Figure 1a). The findings revealed that BHK-21 and DF-1 cells exhibited CPE at 48 hpi, and the cells shrank and gradually shed at 96 hpi. The VERO cell showed cell rounding at 96 hpi, whereas the C6/36 cells did not display significant CPE at 96 h. All four infected cell lines showed replication and proliferation (Figure 1b). The replication of this strain in the BHK-21 and DF-1 cells showed an increasing trend, then decreased within 96 h. Viral titers peaked at 1 × 10^7.88^ PFU/mL in DF-1 cells and at 1 × 10^7.49^ PFU/mL in BHK cells at 60 hpi, in the later stages of infection due to cell death, the viral titer decreased until 96 hpi. In the VERO and C6/36 cells, the viral replication increased with the increasing infection time up to 96 h. At 96 hpi, the viral titers were 1 × 10^5.98^ PFU/mL and 1 × 10^6.52^ PFU/mL in the VERO and C6/36 cells, respectively.

### 3.4. Phylogenetic Analysis

The genome sequences of TMUV-YN2020-20 were obtained by high-throughput sequencing, and the total genome length for this strain was 10,994 nt, containing the 5′-UTR (94 nt), ORF region (10,275 nt/3425 aa), and 3′-UTR (625 nt). The result of Sanger sequencing was consistent with the high-throughput sequencing.

To further investigate the evolutionary relationships of TMUV-YN2020-20, the ORF regions of 8 mosquito-derived and 139 avian-derived TMUVs were examined. TMUV can be formed to 3 clusters: Cluster 1, 2, and 3, which were further classified into several sub-clusters (Cluster 2.1, 2.2, 3.1, and 3.2) (Figure 2a). TMUV-YN2020-20 was located in Cluster 3.2 and closest to YN12193 (KT607936), a mosquito-derived TMUV strain isolated from Yunnan province in 2012. Additionally, Cluster 3.2 included YN12115 (KT607935) (a mosquito-derived strain) and SD14 (MH748542) (an avian-derived strain).

### 3.5. Estimation of Divergent Time of TMUVs

MCC trees were reconstructed based on 148 TMUV E genome sequences, with the estimated time of tMRCAs of these strains shown in the branching nodes. The MCC tree showed three genotypic clusters (Cluster 1, 2, and 3) (Figure 2b) and was consistent with the results of the ML tree (Figure 2a). The tMRCAs of TMUV were estimated in 1924 (95% HPD interval: 1911.2–1936.4), with a mean evolutionary rate estimated at 1.257 × 10^−3^ for the substitution/site/year (95% HPD interval: 1.0586 × 10^−3^ to 1.4716 × 10^−3^). TMUV-YN2020-20 diverged from TMUV-YN12193 (KT607936) in 2008. The Bayesian skyline plot analysis showed the population of TMUV was relatively constant until 2005 (Figure 2c). From 2005 to 2012, the population fluctuated and then levelled off.

### 3.6. Genome Sequence Analysis

The nucleotide and amino acid similarities between this new sequence and other TMUVs in different genomic regions are shown in Table 1. The results showed that TMUV-YN2020-20 and strain YN12193 (KT607936) had the highest similarity in the ORF region, with a 97.7%/99.4% nucleotide/amino acid similarity, and the prM region with a 100% amino acid similarity. The ORF region of this strain showed 94.0%/97.7% and 93.4%/98.3% nucleotide/amino acid similarity to SD14 (MH748542) and YN12115 (KT607935), 87.6%/97.3% and 87.4%/97.2% similarity with TP1906 (MN747003) and NTUC225/20 (MW821486), and 89.7%/97.6% and 88.3%/97.3% similarity with MM1775 (JX477685) and Sitiawan (JX477686).

Additionally, we identified 66 cluster-specific amino acids between Cluster 1, Cluster 2, and Cluster 3 TMUVs, most of which were located in the E protein, followed by NS1, NS2A, and NS5 proteins (Table 2). By analyzing the amino acid variability in ORF sequences between TMUV-YN2020-20 and other TMUVs, we found five unique amino acid variants in TMUV-YN2020-20 located on the E (V358I), NS1 (Y/F/I113L), NS4A (T/A89V), NS4B (D/E/N/C22S), and NS5 (E638G) genes, respectively.

## 4. Discussion

A TMUV strain, TMUV-YN2020-20, was isolated from *Culex tritaeniorhynchus* derived from cattle farms in Yunnan province, China. This strain was replicated in the BHK-21, DF-1, VERO, and C6/36 cells, and had significant CPE in all cells except C6/36. The analysis of the viral genome revealed that it belonged to Cluster 3.2, which was most closely related to the YN12193 isolate at the nucleotide/amino acid level, as well as at the genetic evolutionary level. In addition, this strain evolved five novel amino acid mutations.

TMUV-YN2020-20 does not produce significant CPE in C6/36 cells, and this phenomenon also existed in the study of TMUV-TP1906 strain [32]. However, these two strains could replicate in C6/36 cells. A reasonable explanation is that mosquito-derived TMUV has evolved over a long period to adapt to mosquitoes and mosquito-derived cell lines, and then is transmitted by mosquitoes. *Culex* mosquitoes, such as *Cx. pipiens* and *Cx. tritaeniorhynchus*, are the main vectors of TMUV in nature [27,33]. It has not been determined whether *Aedes* mosquitoes can act as vectors of TMUV; however, in studies on the efficacy of mosquito transmission, it was found that *Aedes albopictus* can be infected with TMUV [34]. Additionally, *A. albopictus* can act as an effective vector of virus transmission when the titer of the infecting virus reaches a certain level, indicating the importance of mosquitoes in TMUV transmission.

Phylogenetic and population dynamic analyses pointed out that Cluster 1 contained avian-derived TMUV strains from Thailand and Malaysia; most of the avian-derived TMUV strains from China were located in Cluster 2, while Cluster 3 contained mosquito-derived TMUV strains. Furthermore, these mosquito-derived strains cluster with the avian-derived strains, indicating that TMUV has the potential for mosquito-to-avian transmission. The Yunnan mosquito-derived TMUVs (YN12115, YN12193, and YN2020) belong to Cluster 3.2 and these Yunnan strains share the most recent common ancestor, suggesting that they have the same evolutionary origin. TMUV is currently thought to survive only in mosquitoes or avian species; however, no cases of the virus infection in diseased ducks were found in Yunnan province until 2019 [35]. Additionally, TMUV was last found in mosquitoes from Yunnan in 2012. We consider that mosquitoes in Yunnan province continue to carry the virus due to the similarity between the mosquito-derived TMUVs found in 2012 and 2020. Furthermore, a conjecture is that TMUV may only circulate within mosquitoes before duck infection, or in a more covert manner in mosquito-duck; nonetheless, either scenario requires mosquitoes to play a key role. In addition, we could not analyze the more precise association between mosquito-derived and duck-derived TMUVs in Yunnan province since the sequences of duck-derived TMUVs are not yet available in the database. However, TMUV isolated from the Shandong avian (duck) was present in Cluster 3.2; therefore, it is likely that these mosquito-derived viruses have the ability to transmit to ducks. Long-term continuous monitoring is necessary to better investigate the circulation of TMUV in natural sources. In brief, the re-isolation of TMUV from mosquitoes in Yunnan province after an interval of 8 years demonstrates the persistence of the virus in this area.

There are two main methods of mosquito-borne virus transmission: one is horizontal transmission, including mosquito-animal (mosquito bite transmission) and mosquito-mosquito (sexual transmission), the other is vertical transmission (ovarian transmission) [36]. Mosquito-animal is the general mode of transmission, and several relevant arboviruses, such as JEV, ZIKV, and DENV, use this mode [37,38]. Mosquito-transmitted TMUV is also considered to rely on this mode [39]. Sexual transmission is a specific type of transmission by which female mosquitoes transmit the virus to male mosquitoes that do not suck blood to expand the ecological range of the virus; ZIKV and others have been found to use this method [40]. Vertical transmission is a mechanism by which viruses can maintain circulation in nature under unfavorable conditions [41]. Members of the family Flavivirus, such as DENV [42] and ZIKV [36], have been found to be transmitted to offspring via eggs in *Aedes aegypti* and *Aedes albopictus*. No study has confirmed that TMUV can be sexually transmitted. In laboratory studies, it was concluded that duck-derived TMUV DK/TH/CU-1 cannot be transmitted vertically in mosquitoes [43]. Whether mosquito-derived TMUV can be transmitted vertically in mosquitoes and its mode of transmission in birds require additional studies.

TMUV-YN2020-20 has five new amino acid mutations (V358I in the E protein, Y/F/I113L in the NS1 protein, T/A89V in the NS4A protein, D/E/N/C22S in NS4B protein, and E638G in the NS5 protein). Amino acid mutation may contribute to the adaptive evolution of the virus to vector [44]. It has been shown that the glycosylation site at amino acid 154 of most flavivirus E proteins can facilitate virus transmission by overcoming the tissue barrier or antiviral response in mosquitoes [45]. Furthermore, amino acid residues at 304 and 367 of the E protein are associated with the virulence and pathogenicity of the virus [46,47]. In addition to the E protein, mutations at amino acid residues 543 and 711 of the NS5 protein affect viral replication and infectivity in the host [48]. The effects of these novel mutations on the function of viral proteins, viral replication, and virulence of the virus require further investigation.

In conclusion, a newly discovered TMUV isolate was analyzed for cellular adaptation and gene sequence characteristics, emphasizing the diversity of TMUV genetic evolution. At this stage, most studies on the pathogenicity of TMUV have focused on Cluster 2, while the remaining genotypes have been less studied, and mosquito-to-avian transmission patterns have not been studied in depth. It is necessary to conduct continuous monitoring of mosquitoes to further investigate the epidemiology of TMUV in Yunnan, China.

## Figures and Tables

**Figure 1 pathogens-12-00490-f001:**
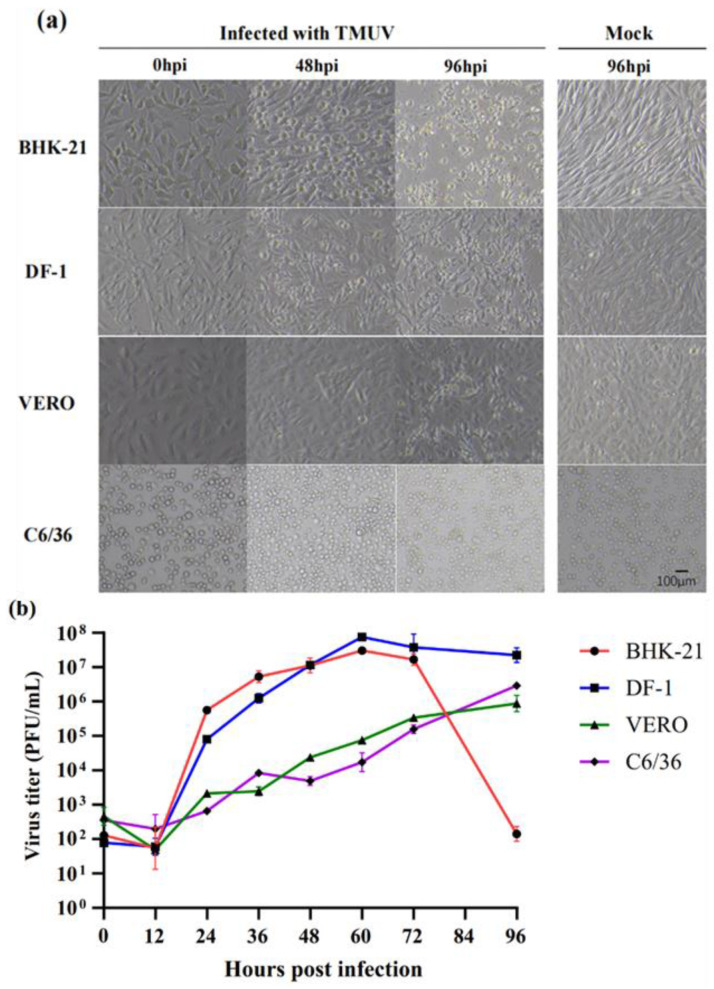
(**a**) Cytopathic changes observed in BHK-21, DF-1, VERO, and C6/36 cells at 0, 48, and 96 hpi. Cells were infected with TMUV-YN2020-20 or mock-infected at an MOI of 0.1 and cell morphological changes were observed using brightfield microscopy; scale bar = 100 μm. (**b**) Viral replication in different cell lines. BHK-21, DF-1, VERO, and C6/36 cells were infected with the virus at 0.1 MOI, and collected at 0, 12, 24, 36, 48, 60, 72, and 96 hpi. The results showed the titer of the virus quantified by standard plaque assay and expressed as PFU/mL. Two biological replicates were conducted for each cell line.

**Figure 2 pathogens-12-00490-f002:**
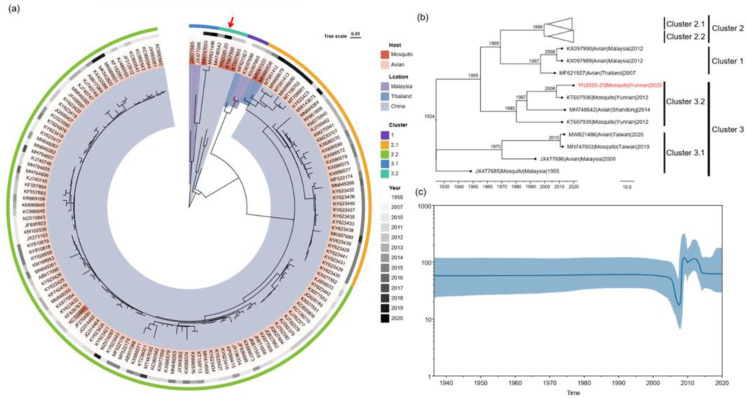
(**a**) ML tree of the ORF region of 148 TMUV strains. (**b**) Bayesian maximum clade confidence (MCC) trees based on 148 TMUV sequence E genes under the GTR + G + I best-fit substitution model. Estimating times to the most recent common ancestors (tMRCAs) of these strains are indicated in the nodes. The scale bar on the bottom shows time in years. The red arrows and the red font indicate the strains YN2020-20 isolated in this article. (**c**) Bayesian skyline plot of TMUV, with the year of isolation of the TMUV on the *x*-axis and the genetic diversity of TMUV shown on the *y*-axis. The 95% highest posterior density (HPD) is shown in blue.

**Table 1 pathogens-12-00490-t001:** Comparison of amino acid and nucleotide similarities between TMUV-YN2020-20 and other TMUVs in genomic sequences.

Genomic Region	Cluster 3							Cluster 1	Cluster 2
	YN12193	SD14	YN12115	MM1775	Sitiawan	TP1906	NTUC225/20	\	\
**ORF**	97.7%	94.0%	93.4%	89.7%	88.3%	87.6%	87.4%	89.8–90.0%	89.0–89.9%
	(99.4%)	(97.7%)	(98.3%)	(97.6%)	(97.3%)	(97.3%)	(97.2%)	(96.5–96.8%)	(95.9–97.6%)
**C**	98.6%	93.9%	96.1%	93.6%	91.7%	91.7%	91.4%	93.1–93.3%	90.6–93.1%
	(99.2%)	(95.8%)	(97.5%)	(96.7%)	(95.8%)	(95.8%)	(95.8%)	(95.0–95.8%)	(91.7–95.0%)
**prM**	97.6%	94.0%	93.2%	91.6%	86.4%	87.2%	87.0%	89.2–90.6%	88.0–89.8%
	(100%)	(96.4%)	(98.8%)	(100%)	(97.0%)	(98.8%)	(98.2%)	(97.6–97.6%)	(95.8–98.2%)
**E**	97.9%	93.4%	94.0%	89.2%	87.3%	86.4%	86.1%	89.1–89.7%	87.6–89.6%
	(99.6%)	(95.2%)	(97.4%)	(97.2%)	(97.0%)	(97.0%)	(96.8%)	(95.6–96.2%)	(94.6–96.8%)
**NS1**	97.1%	94.2%	92.5%	89.4%	87.9%	88.2%	87.8%	89.2–89.7%	86.7–88.8%
	(98.0%)	(98.0%)	(96.9%)	(96.0%)	(94.9%)	(95.7%)	(95.5%)	(94.6–95.5%)	(88.9–94.6%)
**NS2A**	98.1%	92.8%	91.6%	88.5%	89.3%	86.6%	86.6%	87.8–88.0%	86.6–88.7%
	(99.1%)	(96.0%)	(96.0%)	(94.7%)	(94.3%)	(94.3%)	(94.3%)	(92.1–93.8%)	(91.6–93.8%)
**NS2B**	97.5%	94.7%	93.4%	89.3%	89.3%	86.3%	87.5%	90.3–90.6%	88.5–91.3%
	(99.2%)	(99.2%)	(100%)	(96.9%)	(96.9%)	(93.9%)	(96.2%)	(98.5–99.2%)	(95.4–98.5%)
**NS3**	97.3%	94.3%	93.4%	90.6%	89.4%	89.4%	89.3%	90.5–90.9%	89.8–91.0%
	(99.5%)	(99.4%)	(99.2%)	(99.4%)	(98.9%)	(98.7%)	(98.7%)	(98.4–98.5%)	(97.9–98.9%)
**NS4A**	98.1%	92.6%	92.9%	88.9%	87.0%	87.0%	86.2%	88.4–88.9%	84.9–88.1%
	(99.2%)	(96.8%)	(97.6%)	(99.2%)	(97.6%)	(97.6%)	(96.8%)	(95.2–96.0%)	(94.4–97.6%)
**2K**	97.1%	92.8%	94.2%	85.5%	85.5%	87.0%	87.0%	84.1–88.4%	88.4–94.2%
	(100%)	(95.7%)	(100%)	(95.7%)	(95.7%)	(95.7%)	(95.7%)	(100–100%)	(100–100%)
**NS4B**	97.4%	93.4%	92.4%	89.5%	87.1%	86.0%	85.7%	88.5–89.8%	85.6–89.9%
	(99.2%)	(96.9%)	(98.0%)	(97.6%)	(98.0%)	(97.6%)	(97.2%)	(94.9–96.1%)	(91.3–98.0%)
**NS5**	98.0%	94.5%	93.9%	89.6%	88.6%	87.5%	87.2%	90.1–90.5%	89.5–90.9%
	(99.9%)	(99.0%)	(99.2%)	(97.8%)	(98.1%)	(98.1%)	(98.0%)	(97.2–97.9%)	(97.5–98.5%)

Note: Amino acid similarity is indicated in parentheses. GenBank accession number: KT607936 for YN12193, MH748542 for SD14, KT607935 for YN12115, JX477685 for MM1775, JX477686 for Sitiawan, MN747003 for TP1906, and MW821486 for NTUC225/20.

**Table 2 pathogens-12-00490-t002:** Amino acid sites with cluster specificity in Cluster 1, Cluster 2, and Cluster 3.

GENOMIC REGION	SITES	CLUSTER 1	CLUSTER 2	CLUSTER 3 ^2^
C	34	M (100%) ^1^	T (100%)	T (100%)
	40	K (100%)	R (97.81%); T (2.19%)	R (100%)
	61	K (100%)	R (100%)	K (100%)
	76	N (100%)	N (98.54%); D (1.46%)	S (100%)
E	2	S (100%)	S (100%)	N (100%)
	52	D (100%)	E (100%)	E (100%)
	72	S (100%)	S (100%)	P (100%)
	83	S (100%)	P (100%)	P (100%)
	89	D (100%)	E (98.54%); G (0.73%); K (0.73%)	D (87.50%) *; N (12.50%)
	157	A (100%)	A (97.81%); V (2.19%)	V (100%)
	180	L (100%)	M (100%)	L (100%)
	185	S (100%)	T (98.54%); A (1.46%)	A (62.50%) *; S (25.00%); T (12.50%)
	236	K (100%)	K (100%)	R (100%)
	312	A (100%)	V (95.62%); A (4.38%)	A (100%)
	332	S (100%)	T (97.81%); A (1.46%); I (0.73%)	S (100%)
	358	V (100%)	V (100%)	V (87.50%%); I (12.50%) **
M	22	L (100%)	L (99.27%); N (0.73%)	M (87.50%) *; T (12.50%)
	24	I (100%)	V (100%)	I (87.50%) *; V (12.50%)
	29	T (100%)	A (100%)	T (100%)
	60	H (100%)	Y (97.08%); H (2.92%)	Y (100%)
	115	I (100%)	T (100%)	T (100%)
	150	S (100%)	S (99.27%); G (0.73%)	G (87.50%) *; S (12.50%)
NS1	2	V (100%)	T (100%)	M (100%)
	21	V (100%)	V (100%)	I (87.50%) *; V (12.50%)
	41	R (100%)	R (99.27%); K (0.73%)	K (100%)
	51	K (100%)	E (89.78%); V (10.22%)	E (100%)
	83	G (100%)	A (100%)	G (50.00%) *; S (50.00%)
	99	R (100%)	K (99.27%); R (0.73%)	R (100%)
	108	E (100%)	D (98.54%); G (1.46%)	E (100%)
	113	F (100%)	Y (100%)	F (75.00%); I (12.50%); L (12.50%) **
	139	K (100%)	K (100%)	R (100%)
	274	K (100%)	V (98.54%); A (1.46%)	K (100%)
	351	M (100%)	M (100%)	V (100%)
NS2A	31	P (100%)	P (100%)	S (100%)
	35	S (100%)	S (100%)	P (100%)
	53	E (100%)	D (100%)	D (100%)
	68	R (100%)	S (98.54%); N (1.46%)	S (100%)
	122	D (100%)	N (100%)	N (100%)
	123	I (100%)	M (97.81%); I (2.19%)	M (100%)
	150	L (100%)	F (100%)	L (100%)
	157	S (100%)	L (98.54%); V (1.46%)	S (100%)
	187	S (100%)	S (100%)	N (100%)
NS2B	95	L (100%)	L (100%)	F (100%)
NS3	15	R (100%)	R (100%)	K (100%)
	258	I (100%)	I (100%)	V (100%)
	324	E (100%)	E (99.27%); D (0.73%)	D (100%)
	591	I (100%)	I (98.54%); T (0.73%); V (0.73%)	T (100%)
NS4A	70	L (100%)	F (99.27%); L (0.73%)	F (100%)
	89	A (100%)	T (100%)	A (87.50%); V (12.50%) **
	96	M (100%)	I (100%)	V (100%)
	110	I (66.67%); F (33.33%)	V (99.27%); I (0.73%)	I (100%)
NS4B	14	A (100%)	S (100%)	S (100%)
	22	D (100%)	D (89.78%); N (1.46%); E (8.76%)	D (50.00%); N (25.00%); C (12.50%); S (12.50%) **;
	60	I (100%)	I (100%)	V (100%)
	118	V (100%)	I (100%)	I (100%)
	183	M (100%)	V (99.27%); M (0.73%)	V (75.00%) *; A (25.00%)
NS5	133	Y (100%)	H (98.54%); Q (1.46%)	H (100%)
	150	A (100%)	A (100%)	S (100%)
	188	M (100%)	M (100%)	T (100%)
	232	G (100%)	S (100%)	S (87.50%) *; T (12.50%)
	246	R (100%)	K (99.27%); R (0.73%)	R (100%)
	288	K (100%)	R (100%)	K (100%)
	388	S (100%)	G (99.27%); S (0.73%)	G (100%)
	390	D (100%)	N (100%)	N (100%)
	448	R (100%)	K (100%)	K (100%)
	638	E (100%)	E (100%)	E (87.50%); G (12.50%) **
	790	I (100%)	V (100%)	V (100%)

^1^ The proportion of amino acid sites in Cluster 1, Cluster 2, or Cluster 3. ^2^ The amino acid sites of YN2020-20 are contained in Cluster 3. * The amino acid of TMUV-YN2020-20 at this polymorphic site; ** The unique amino acid mutation of TMUV-YN2020-20.

## Data Availability

All the data generated during the current study are included in the manuscript and/or the Appendix A. Additional data related to this article may be requested from the corresponding authors.

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
