# Peer review of "Persistence of Tembusu Virus in Culex tritaeniorhynchus in Yunnan Province, China"

_pathogens, 2023, doi:10.3390/pathogens12030490_

Round 1

Reviewer 1 Report

This study by Hu et al. reports the isolation of TMUV from Culex tritaeniorhynchus mosquitoes in Yunnan province, China, in 2020. The growth characteristics in different cell lines were compared. The sequence of the isolated virus was determined, and amino acid and nucleotide homologies to other TMUV isolates were calculated. A phylogeny of known TMUV sequences was generated, and the tMRCA of TMUVs and the divergence point of the new virus isolate from its closest relative estimated. 

Overall, I found the study really interesting. The manuscript needs some editing for language/grammar, some of the content and the discussion. I would kindly ask the authors to consider the following comments before publication. Thank you.

Abstract: The term "avian diseases" (plural) is not specific enough. Causes disease in birds?; Please mention the virus family here. 

Line 47: Please edit grammar: "The infectious disease of ducks infected with TMUV..."

Line 56: dengue virus is spelled with a small d

Line 91: BHK cells are derived from golden hamster (Mesocricetus auratus)

Line 95: Species names in italics, please

Line 121: "a few days of incubation" - please be specific

Line 172 onwards (paragraph 3.2): Please mention that you isolated virus from 1 mosquito out of ~5000 (or whatever the finding was)

Figure 1a: why not leave Mock control cells for 96 hrs?

197: should read "cell line" instead of "cell lines"

In paragraph 3.4 you use the ORF sequence of TMUVYN2020-20 for phylogenetic analyses but only mention in paragraph 3.6 that you have obtained the genomic sequence by high-throughput sequencing. This needs to be brought in a logical order. Perhaps move the first sentence of p3.6 to p3.4?

Paragraph 3.5: Please spell out MCC, ML and tMRCA for non-expert readers to better understand what you are talking about. Rewrite the following sentence as: "MCC tree showed three genotypic clusters (Cluster 1, 2 and 3) (Figure 2b) and was consistent with the result of (the) ML tree (Figure 2a)." - i.e. figure references to be placed at the end of each argument.

Line 222: it should read "levelled" instead of "level"

Paragraph 3.6: Mention that you compared sequences of cluster 3 in Figures 2a and 2b. Isolates Sitiawan, TP1906, NTUC22 5/20 are not mentioned in the text and no accession number that would allow finding them in Figures 2a and 2b is given. Please add. 

Table 2: It remains unclear what the difference between black bold ("different from TMUVs") and red bold ("unique sites") is. 

Line 255: "which is same" - please fix the grammar

Line 286: "demonstrating" should read "demonstrates"

Line 302: I am really not sure that if you isolate an arbovirus from a mosquito vs an animal, the amino acid sequence changes you find indicate some sort of adaptation to where you isolated the virus from. Arboviruses have a comparatively short replication time in both mosquitoes and animal hosts (several days only). There are strong selection barriers to mutations in mosquitoes and there is hardly any evolution happening in the mammalian host. Indeed, mutations are limited by the two-host nature of arboviruses. How do you know that your virus isolate has not cycled many times between Culex and birds before you eventually isolated it? You see some clustering of mosquito isolates with duck isolates in cluster 3. Is it possible that mosquitoes simply have not been sampled in other regions and this is why you have no mosquito sequences associating with clusters 1 or 2? I honestly would not want to overinterpret the genotype-host relationship too much. 

Line 315: "investigated" should read "investigation"

Reviewer 2 Report

This manuscript describes the isolation and characterization of a Tembusu virus strain form Culex tritaeniorhynchus.  The procedures and analysis are appropriate and fairly well described.  The results; however, seem some what lacking on describing the overall project.  Only one isolate was fully characterized.  Was there only one isolate?  Much was made about the changes in the virus but the virus was isolated after 3 passages.  What about passaging affects on virus genetics.  It also wasn’t stated which cells the virus was isolated from.  There is discussion on the lack of CPE in C6/36 cells.  This is standard for arboviruses.  Much of the manuscript seems to jump from one idea to the next.  Would recommend re-writing to a more cohesive story.  The isolation and characterization of a Tembusu virus strain after eight years of no activity is worthy of reporting as a short note.  Was this isolation because no one was looking or is there and environmental change – outbreak that led to the discovery.

Specific Comments

Ln 100-105: Were three blind passages always used or did only one isolate result after three blind passages?  

Ln 177-179:  This paragraph states positive cultures.  How many?  Only one strain sequenced.  What was the infection rate in the collected mosquitoes?x

Ln 274:  replace “latest” with “last” 

Ln 288-290:  States 2 modes then lists 3?  How is this paragraph relevant to the rest of the manuscript.

Figure 1A  is not needed.  Fairly standard results for arboviruses.

Figure 1B:  Why the drop off for BHK-21 cells?  This was not discussed in the text.

Figure 2C:  This is the most interesting analysis in the paper but is not discussed in the text.  Why the blip in instability ~2010?  Is this just do to data availability?

Round 2

Reviewer 1 Report

Dear authors, thank you for thoroughly addressing my comments.  

Re the spelling of dengue virus: it is not spelled with a capital d because "dengue" is not a place name, unlike in JEV and ZIKV, who have their origin in Japan and the Zika Forest in Uganda, respectively. Equally yellow fever virus is also spelled with a small y for the same reason (not a place name). 
